# Comparison of the New Refrigerant R1336mzz(E) with R1234ze(E) as an Alternative to R134a for Use in Heat Pumps

**Jan Drofenik \*, Danijela Urbancl and Darko Goričanec**

Faculty of Chemistry and Chemical Engineering, University of Maribor, Smetanova 17, 2000 Maribor, Slovenia; danijela.urbancl@um.si (D.U.); darko.goricanec@um.si (D.G.)

\* Correspondence: jan.drofenik@um.si

**Abstract:** R134a is currently the most widely used refrigerant, whose problem is the high value of the global warming potential, and which will have to be replaced in the near future. Thus far, R1234ze(E) has proven to be the most suitable alternative, but it is slightly flammable. Recently, R1336mzz(E) has emerged as a possible alternative. During the research, the mentioned refrigerants were compared with simulations using the Aspen Plus software package in the case of using groundwater as a heat source. It was found that R1336mzz(E) could be a suitable alternative to R134a since the highest value of coefficient of performance was obtained using it. However, it must be superheated with an internal heat exchanger. The problem with using an internal heat exchanger is that the superheating of the refrigerant vapors affects the isentropic efficiency of the heat pump compressor negatively and, consequently, lowers the COP value of the heat pump. It has been shown that a one percent decrease in isentropic efficiency results in a one percent decrease in the COP value.

**Keywords:** heat pump; internal heat exchanger; HFO; HFC alternative; GWP; R134a; R1234ze(E); R1336mzz(E)

## 1. Introduction

Almost all industries require heat for their operations. Even today, most thermal energy in the EU is derived from fossil fuels. These are not available in unlimited quantities. Therefore, it is necessary to look for ways to use them more efficiently and gradually replace them with renewable energy sources. In its 2012 Directive 2012/27/EU, the European Union committed to its members to use at least 20% renewable energy sources for heat production by 2020 and to achieve 20% energy efficiency by 2020 [1]. In 2018, the strategy was amended to 2030. The complementary strategy 2018/2002 states that the energy efficiency of EU member states must be at least 32.5% by 2030 [1].

The demand for greater energy efficiency will increase the use of heat pumps that use renewable heat sources to generate heat, but the refrigerants that transfer heat through the system continue to cause greenhouse gas emissions. Currently, hydrofluorocarbons (further HFC), which are third-generation refrigerants, are the most used refrigerants. Although HFC are not as dangerous to ozone layer depletion as their precursors CFC and HCFC, they have a high global warming potential (further GWP) [2]. One of the most used refrigerants currently, R134a, has a GWP of about 1300 kg of greenhouse gas (further GHG) emissions in $CO_2$ equivalents per kilogram of refrigerant [3]. According to EU Regulation 517/2014, the use of refrigerants with a GWP higher than 150 in domestic refrigerators and freezers was prohibited from January 1, 2015 [2,4]. As of January 1, 2017, air conditioning systems, refrigeration equipment and heat pumps filled with HFC as refrigerants can no longer be sold unless special exemptions apply [4,5]. The EU has set a target to reduce emissions of fluorinated GHG by 66% by 2030 compared to 2010 [5]. Hydrofluoroolefins (hereafter HFO), which have a much lower GWP value, have emerged as a possible alternative to HFC.

The new refrigerants must be non-toxic and non-flammable. They must have an ODP value of 0, a low GWP value and a short atmospheric lifetime [6]. Non-traditional HFC

refrigerants with low GWP values have been tested as alternatives to traditional HFC refrigerants. Cabello et al. compared the feasibility of using HFC R152a as an alternative to R134a. Although even higher coefficients of performance (further *COP*) values were obtained with R152a than with R134a, R152a is not a good alternative to R134a because it is flammable [6]. Binary blends of HFC and HFO have also been tested as an alternative to HFC. Such blends could improve the disadvantages of HFO (flammability) and HFC (high GWP) [7]. Mota-Babiloni et al. investigated the possibility of using a blend of R1234ze(E) and R134a, called R450A, as a replacement for R134a. Although this mixture is non-flammable and was found to be an effective substitute, the GWP value is still high compared to pure HFO (547) [7]. Mateu-Roya et al. investigated R515B, a mixture of R1234ze(E) (91.1 wt%) and R227ea (8.9 wt%) as a possible replacement for R134a and compared it with pure R1234ze(E). R515B is non-flammable, and its GWP value is 299, which is much lower than the GWP value of R450A. R1234ze(E) and R515B have been shown to be similarly effective as replacements for R134a. The main advantage of R515B over R1234ze(E) is non-flammability, but the GWP value of R515B is still higher than that approved for use in domestic refrigerators and freezers [8]. The first HFO tested as an alternative to R134a was R1234yf. Shortly thereafter, its structural isomer R1234ze(E) was also tested as an alternative to R134a [3]. Yataganbaba et al. found that the efficiency parameters were slightly lower when using R1234yf than when using R134a, but R1234yf is still a good alternative to R134a given its much lower GWP value [9]. When R1234ze(E) was used, the efficiency parameters were almost identical to the efficiency parameters when R134a was used [9]. Although R1234yf and R1234ze(E) are classified in the same ASHRAE safety class, R1234ze(E) was found to be less flammable than R1234yf [7]. Mota-Babiloni et al. found that the flammability of R1234ze(E) is not critical, and that it is not flammable under normal operation and low humidity [3]. Based on these results, it can be concluded that R1234(E) is currently the most suitable HFO to replace R134a.

In 2020, Sakoda et al. studied the values of PvT parameters, the values of vapor pressures, the values of saturation densities and the values of the critical parameters for R1336mzz(E), which is emerging as a new alternative to HFC [10]. The main advantage of R1336mzz(E) is that it is one of the few non-flammable HFO [10].

The aim of the study is to compare the efficiency of the new refrigerant R1336mzz(E) with the currently most widely used refrigerant R134a and its most suitable environmentally friendly substitute R1234ze(E) by comparing the performance of the refrigerants in different heat pump systems when groundwater is used as the heat source.

## 2. Materials and Methods

A heat pump is a device for transferring heat in the opposite direction to the direction of the spontaneous heat transfer. The most used low-temperature heat sources are:

- geothermal water up to a temperature of 55 °C;
- heat from flue gases;
- low-temperature sources from industrial processes;
- excess heat from cooling systems;
- heat of groundwater, lakes, seawater or rivers.

By using a heat pump, the desired medium or space is heated or cooled. The main advantages of using a heat pump for cooling and heating are the significantly lower cost of electricity consumption and lower GHG emissions into the environment. Simulations of the operation of two different heat pump systems were carried out using the Aspen Plus software package.

### 2.1. Refrigerants

The heat pump takes advantage of changes in the physical state of the refrigerant, which means that a much greater amount of heat can be exchanged between the refrigerant and the medium than if the change in physical state did not occur. The refrigerant must be efficient and harmless to the environment and humans. The harmlessness of a refrigerant

is measured by the ODP and GWP parameters. Refrigerants are classified into ASHRAE safety classes that can be used to compare refrigerants in terms of flammability and toxicity. Simulations were performed with three different refrigerants. The refrigerant 1,1,1,2-tetrafluoroethane, designated as R134a, belongs to the group of fluorinated hydrocarbons that do not contain chlorine atoms, and whose ODP value is 0. The problem with HFC is their high GWP values, which is why they must be phased out in the future [4]. An effective alternative to fluorinated hydrocarbons are HFO, whose ODP values are also 0, and whose GWP values are many times lower compared to HFC. HFO are cost and energy efficient. In the simulations, trans-1,3,3,3-tetrafluoropropene, designated R1234ze(E), and trans-1,1,1,4,4,4-hexafluoro-2-butene, designated R1336mzz(E), were used as refrigerants. Table 1 shows the properties of the refrigerants used in the simulations.

**Table 1.** Comparison of the properties of refrigerants [3,11–17].

| Refrigerant | R134a | R1234ze(E) | R1336mzz(E) |
|---|---|---|---|
| Chemical name | 1,1,1,2-tetrafluoroethane | trans-1,3,3,3-tetrafluoropropene | trans-1,1,1,4,4,4-hexafluoro-2-butene |
| Family | HFC | HFO | HFO |
| Structural formula |  |  |  |
| Molecular weight (g/mol) | 102.03 | 114.04 | 164.05 |
| * NBP (°C) | −26.3 | −19.0 | 7.5 |
| Critical temperature (°C) | 101.1 | 109.4 | 137.7 |
| Critical pressure (MPa) | 4.06 | 3.63 | 3.15 |
| Atmospheric lifetime | 13 years | 18 days | Short |
| ASHRAE safety classification | A1 | A2L | A1 |
| Flammability | Non-flammable | Lower flammability | Non-flammable |
| Toxicity | Lower toxicity | Lower toxicity | Lower toxicity |
| ODP | 0 | 0 | 0 |
| GWP (kg $CO_2$ eq/kg of refrigerant) | 1300 | 4 | 18 |

* NBP stands for normal boiling point.

The A symbol in the ASHRAE safety class means that the refrigerant is of lower toxicity [15]. The designation of all three refrigerants used in the simulations begins with the letter A, indicating low toxicity. The second part of the ASHRAE safety class label classifies the refrigerant according to its flammability. If the second part of the label contains the number 1, it means the refrigerant is not flammable. The number 2 in the second part of the safety class 2 means that the refrigerant is of lower flammability. If the letter L is next to the number 2, it means that the refrigerant is of low flammability [15].

Temperature–Entropy (Further T–S) Diagrams for Different Refrigerants

The T–S curves for each of the refrigerants were generated using the Model Sensitivity Tool in the Aspen Plus software package. The entropy changes at different temperatures were calculated for each refrigerant. This varied in the range from −5 °C to the critical temperature of each refrigerant in 1 °C increments. Entropy changes were calculated at all temperatures when all refrigerants were in the liquid phase and when all refrigerants were in the vapor phase.

### 2.2. Simulation of Heat Pump Operation

The basic heat pump consists of a compressor, a condenser, an expansion valve and an evaporator. The configuration of the compression heat pump is shown in Figure 1.

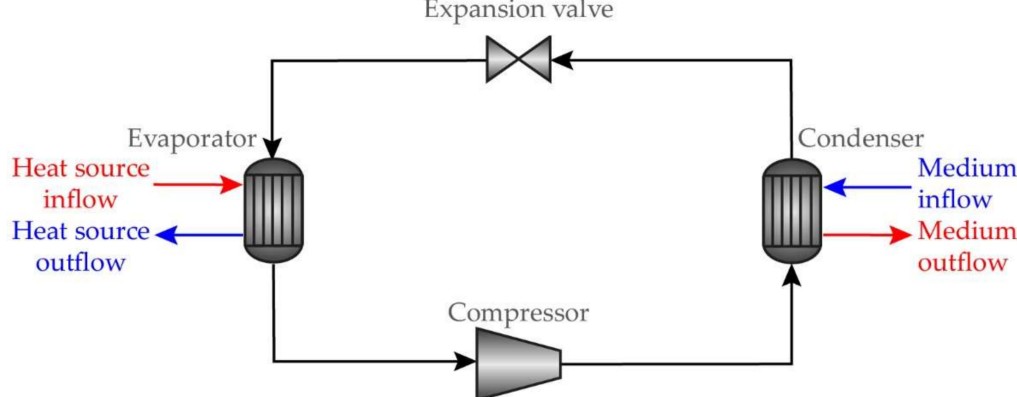

**Figure 1.** Configuration of compression heat pump.

### 2.2.1. Compressor

An isentropic compressor type was used to simulate heat pump operation. The mechanical efficiency of the compressor was estimated to be 95%, and the isentropic efficiency of the compressor was estimated to be 80%. The simulations were performed at a minimum discharge pressure that ensures a sufficiently high temperature of the refrigerant vapors leaving the compressor to avoid a temperature crossover in the condenser.

### 2.2.2. Condenser

The condenser was dimensioned as a countercurrent heat exchanger in the simulation. The minimum temperature approach was 2 °C, which means that, in practice, a plate heat exchanger must be used.

In the condenser, the refrigerant gave heat to the water. The water inlet temperature was 35 °C. The water was heated to three different temperatures, 45, 55 and 65 °C. The flow rate of the heated water was maximized through the condenser.

### 2.2.3. Expansion Valve

The outlet pressure was determined at the expansion valve. This was set for each refrigerant such that the temperatures of the vapors leaving the compressor were equal to 1 °C. Table 2 shows the values of the specific outlet pressures for all three refrigerants ($p_{EV,A}$).

**Table 2.** Expansion valve outlet pressure for different refrigerants.

| Refrigerant | $p_{EV,A}$ (bar) |
|:---:|:---:|
| R134a | 3.025 |
| R1234ze(E) | 2.206 |
| R1336mzz(E) | 0.776 |

### 2.2.4. Evaporator

The evaporator was dimensioned as a countercurrent heat exchanger in the simulation. The minimum temperature approach was 2 °C, which means that, in practice, a plate heat exchanger must be used.

The heat source for the evaporation of the refrigerant was water with an inlet temperature of 10 °C. The refrigerant flowed into the evaporator at a temperature of 1 °C and out of it at a temperature of 8 °C.

### 2.2.5. Heat Pump with Internal Heat Exchanger (Further IHE)

It was useful to simulate a heat pump with an IHE but only in the case where the refrigerant R1336mzz(E) was used. The condensed refrigerant is fed into an IHE before expansion at the expansion valve, where it gives off heat to the refrigerant vapors, causing them to superheat.

The IHE was dimensioned in the simulation as a countercurrent heat exchanger. The minimum temperature approach was 2 °C, which means that, in practice, a plate heat exchanger must be used.

The configuration of the compression heat pump with an IHE is shown in Figure 2.

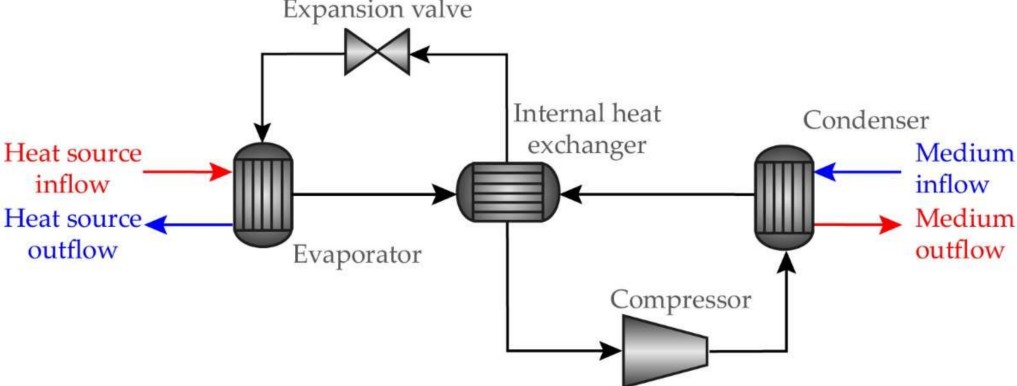

**Figure 2.** Configuration of a compression heat pump with an IHE.

Refrigerant vapors superheat when the IHE is used. Superheating refrigerant vapors before they enter the compressor reduces their density, allowing fewer vapors to enter the compressor at one time. Superheating refrigerant vapors reduces the isentropic and volumetric efficiency of the compressor significantly. The effect of refrigerant vapors superheating on compression losses remains relatively unexplored to date [18].

The calculation of the isentropic efficiency of a compressor $\eta_{isen}$ is shown in Equation (1) [19].

$$\eta_{isen} = \frac{W_{isen}}{W_{actual}}, \tag{1}$$

where $W_{isen}$ stands for the work of the isentropic compression process and $W_{actual}$ for the work of the real compression process [19].

In the IHE heat pump simulations, the isentropic efficiency of the compressor varied in 5% increments between 60% and 80%. Eighty percent represented the isentropic efficiency of the compressor in simulations with the refrigerants R134a and R1234ze(E). In these simulations, the refrigerant was not superheated further in the IHE. The superheating of the refrigerant vapors causes a drop in the isentropic efficiency of the compressor [18]. How the performance parameters changed when the isentropic efficiency of the compressor decreased due to the superheating of the vapors was investigated.

The refrigerant flow in the compression heat pump simulations and in the heat pump simulations with IHE was 500 kg/h.

### 2.3. Simulation of a Series of Heat Pumps

The article compares the COP values of the heat pump with the COP values of the series of two heat pumps when the water was heated to 55 °C in the case when the heat pump condensers were connected together, and in the case when they were connected together and the heat pump evaporators were also connected together. The COP values of the heat pump when the water was heated to 65 °C was also compared with the two possible series of heat pumps. The structures of the heat pumps connected in series for the case when the water was heated to 65 °C are shown in Figures 3 and 4.

Figure 3 shows a diagram of a series of heat pumps with connected condensers. Groundwater at a temperature of 10 °C flows separately into the evaporator of all three heat pumps as a heat source. The heat pumps are connected in series by the flow of the heated medium. This is heated by 12 °C in the condenser of the first heat pump, then by another 10 °C in the condenser of the second heat pump, and finally by another 8 °C in the condenser of the third heat pump. The heated water leaves the series of the three connected heat pumps at a temperature of 65 °C.

Figure 4 shows the structure of a series of heat pumps in which the evaporators are also interconnected. The first evaporator contains groundwater at a temperature of 10 °C, which is cooled in the condenser of the first heat pump. The outflow from the evaporator of the first heat pump is then directed to the evaporator of the second heat pump, where it is cooled further. The groundwater outflow from the evaporator of the second heat pump is then directed to the evaporator of the third heat pump, where it is cooled to the final temperature at which it leaves the system.

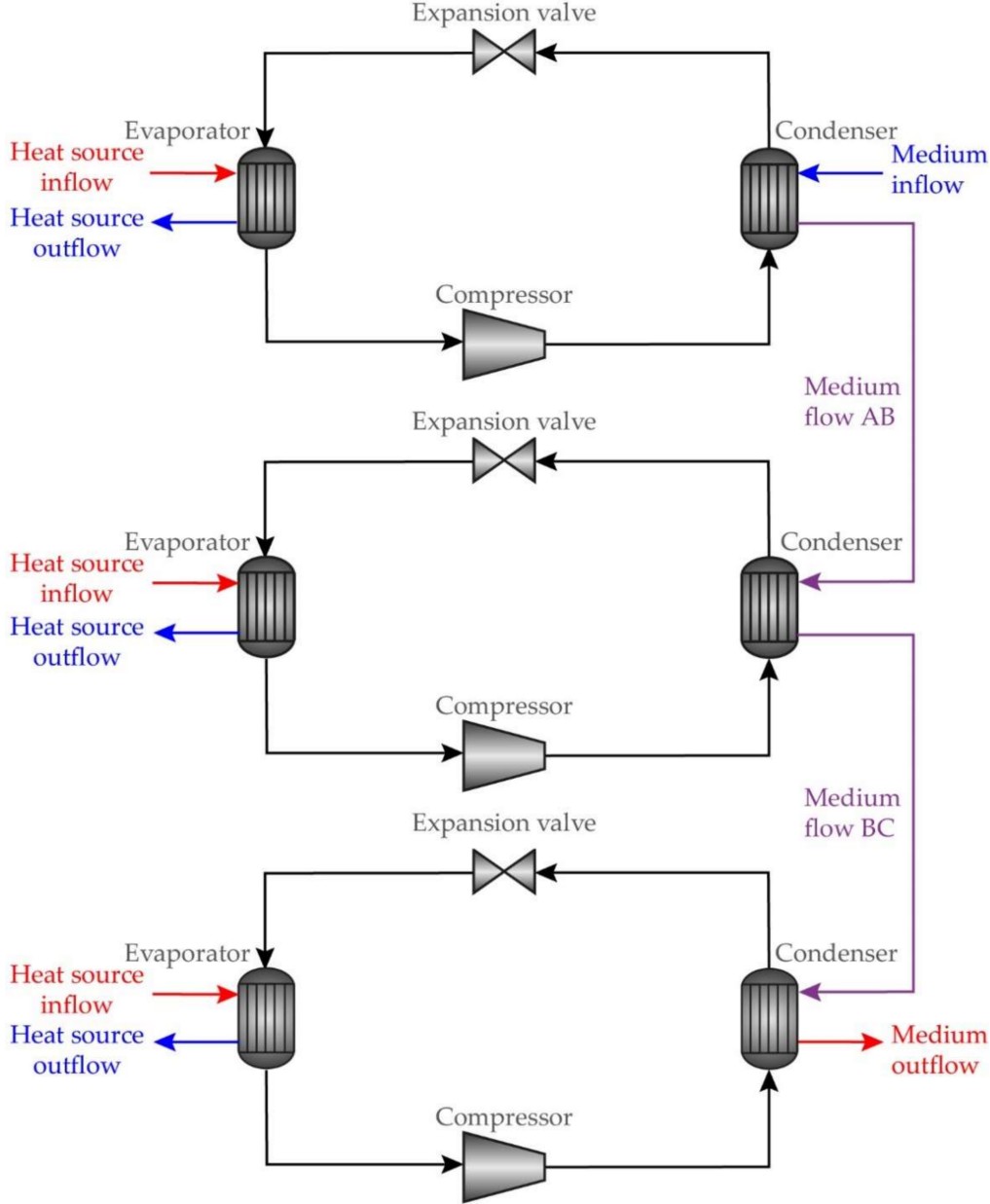

**Figure 3.** Series of heat pumps with connected condensers.

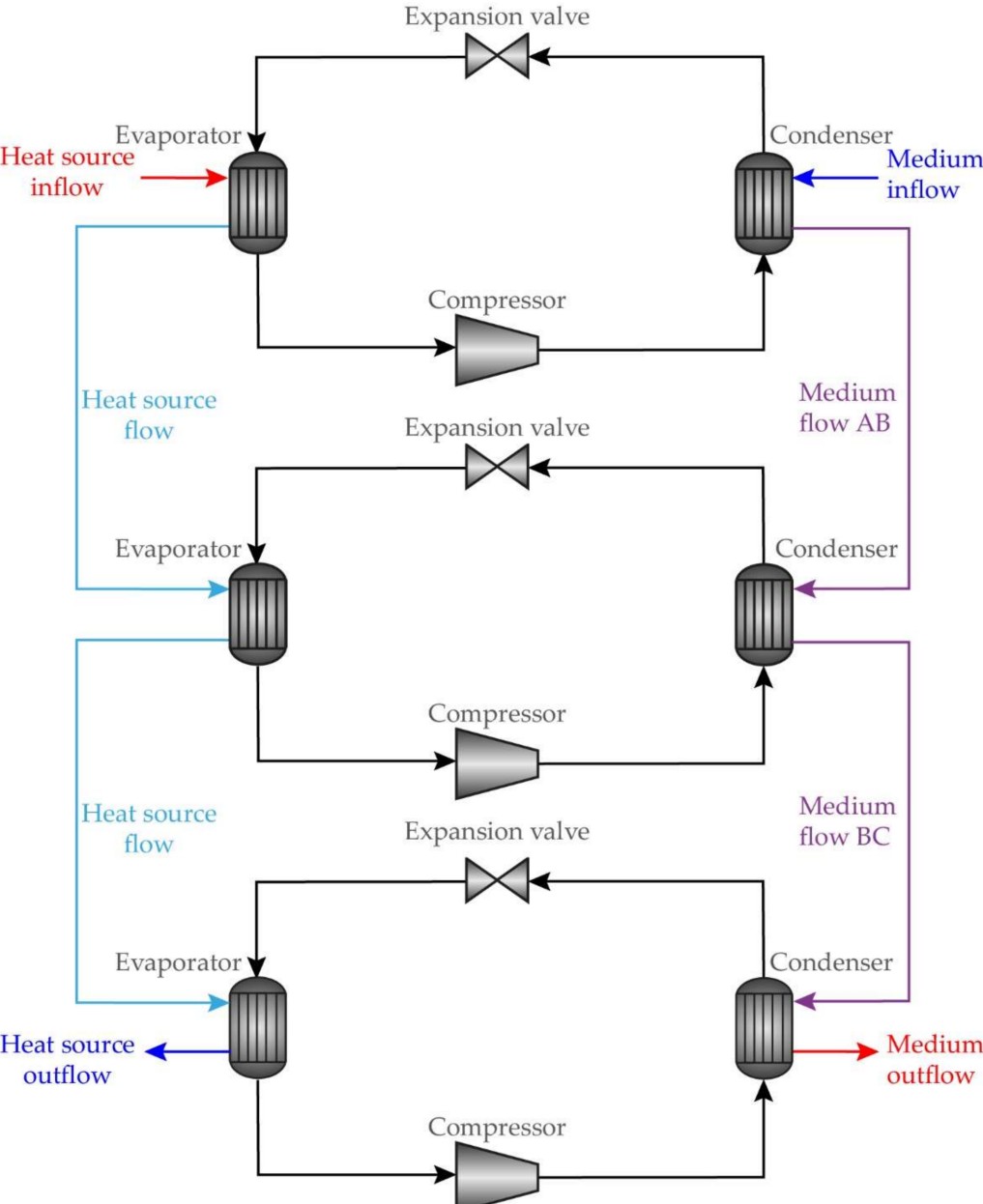

**Figure 4.** Series of heat pumps with connected condensers and evaporators.

## 3. Results and Discussion

When simulating a heat pump in which heat is transferred by the refrigerant R1336mzz(E), it was found that, to use groundwater with a temperature of 10 °C as a heat source, an internal heat exchanger must be added; otherwise, condensation of R1336mzz(E) vapors occurs in the heat pump compressor.

### 3.1. Problems with the Use of R1336mzz(E) in a Basic Heat Pump

During the simulation of the basic heat pump, it was found that, for both HFO refrigerants, there was the problem that the vapors entering the compressor began to condense in the compressor below a certain temperature, which, in practice, leads to the failure of the compressor. This phenomenon occurs because the vapors of the refrigerant are saturated. In the T–S diagram, the compression curve intersects the T–S curve at this error. This error is shown in Figure 5 below for the case where the water is heated to 45 °C

and the vapors of R1336mzz(E) enter the compressor at a temperature of 8 °C, with the discharge pressure of the compressor set at 3.7 bar.

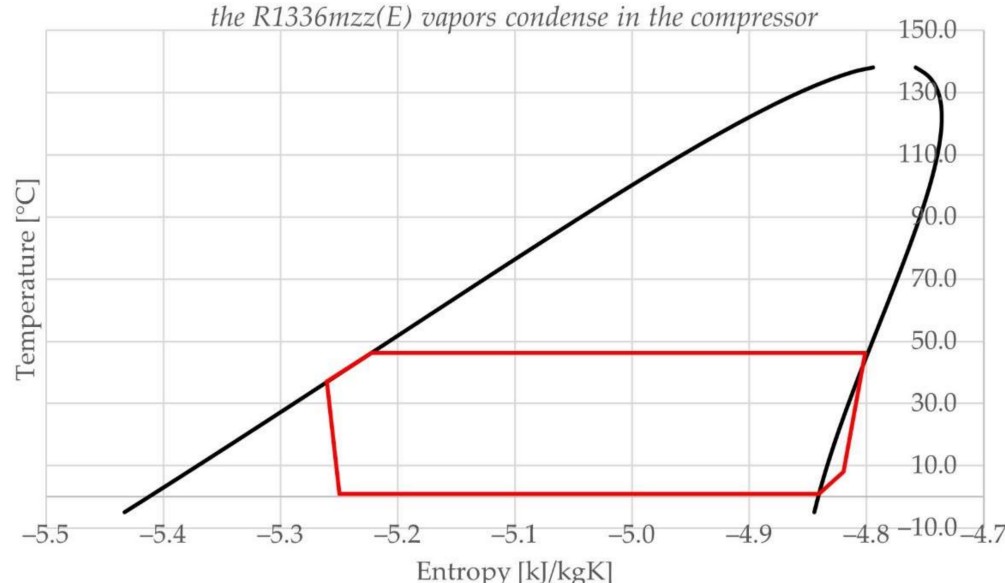

**Figure 5.** Error that occurs when the refrigerant vapor temperature at the compressor inlet is too low.

Table 3 shows the minimum values of temperature ($T_{1,A,MIN}$) of refrigerant vapors at the compressor inlet to prevent condensation in the compressor, and the compressor discharge pressure setting ($p_{COMP,A}$) for both HFO refrigerants for different heated water outflow temperatures.

**Table 3.** Minimum values for the refrigerant vapor temperature when entering the compressor with the specified discharge pressure.

| Refrigerant | Temperature of the Heated Water | 45 °C | 55 °C | 65 °C |
|---|---|---|---|---|
| R1234ze(E) $p_{EV,A}$ = 2.21 bar | $T_{1,A,MIN}$ $p_{COMP,A}$ | 4 °C 8.6 bar | 4.7 °C 11.1 bar | 5.1 °C 14.9 bar |
| R1336mzz(E) $p_{EV,A}$ = 0.78 bar | $T_{1,A,MIN}$ $p_{COMP,A}$ | 14.3 °C 3.7 bar | 17.6 °C 4.9 bar | 21.2 °C 6.4 bar |

The minimum discharge compressor pressures for each refrigerant were determined based on the desired outflow temperature of the heated water. At these pressures, the vapor temperatures at the inlet to the compressor at which the R1234ze(E) began to condense therein ranged from 4 to 5.1 °C and were lower than 8 °C, which was the maximum possible vapor temperature at the outlet of the evaporator for the selected heat source. Therefore, when R1234ze(E) was used, it was not necessary to install an IHE in the heat pump system. When the refrigerant R1336mzz(E) refrigerant was used, the vapor temperatures at which the refrigerant was started ranged from 14.3 to 21.2 °C. Since the minimum outlet temperatures for R1336mzz(E) from the evaporator are higher than the inlet temperature of the heat source, it is not possible to use R1336mzz(E) in the basic heat pump because of a temperature crossover in the evaporator, as shown in Figure 6.

### 3.2. Use of R1336mzz(E) in a Heat Pump with an IHE

In the case of using R1336mzz(E) as the refrigerant, groundwater can be used as the heat source if the IHE is installed in the heat pump. The configuration of such a heat pump is shown in Figure 2. The superheat temperature of the evaporated R1336mzz(E) in the IHE was equal to the difference between the minimum temperature of the refrigerant vapor,

given in Table 3, and the outlet temperature of the refrigerant vapor at the outlet of the evaporator. The temperatures at which the vapor of R1336mzz(E) is superheated in the IHE are related directly to the outlet temperature of the heated water from the condenser.

Figure 7 shows the dependence of the temperature at which the vapor of the R1336mzz(E) must be superheated in the IHE on the temperature of the heated water at the outlet of the condenser. The red curve shows the minimum temperature of the saturated vapors entering the compressor, at which condensation does not yet occur in the compressor. The blue curve shows the temperature of the vapor at the outlet of the evaporator of the heat pump. The higher the desired temperature of the heated water, the more the vapor R1336mzz(E) must be superheated before it enters the compressor.

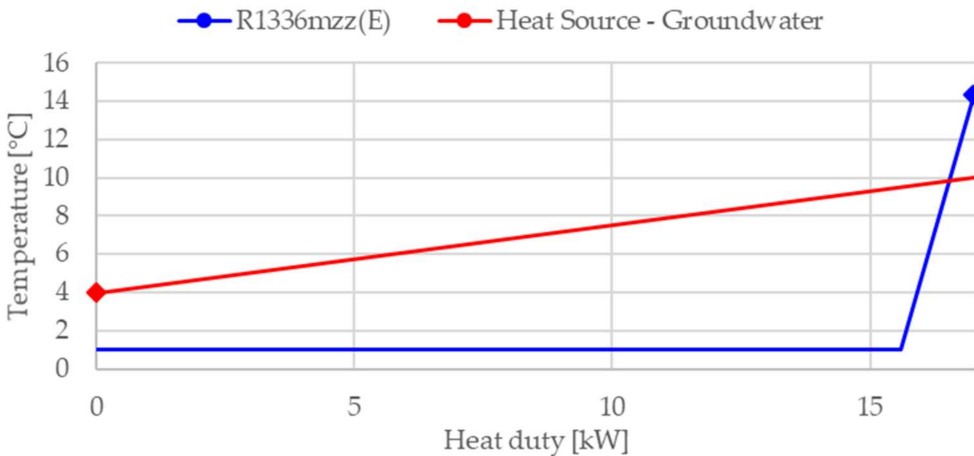

**Figure 6.** Temperature crossover, which occurs in the evaporator when R1336mzz(E) is used in a basic heat pump.

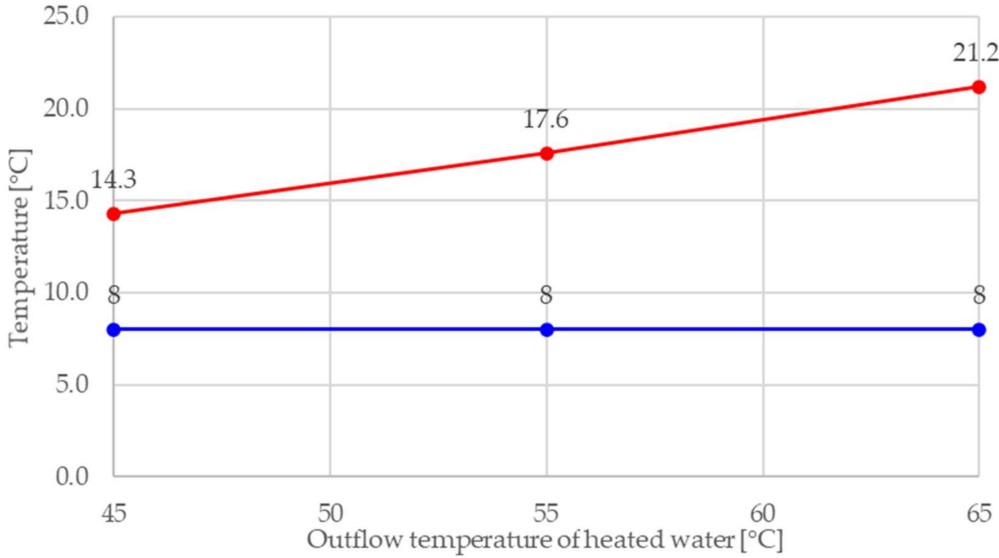

**Figure 7.** Superheating of R1336mzz(E) in the IHE.

Figure 8 shows a T–S diagram of an IHE heat pump with R1336mzz(E) as the refrigerant. The light blue line shows the processes that take place in the process units that make up the basic heat pump. The yellow line, however, shows the changes that take place in the IHE, which is additionally built into the heat pump. The line between points 1 and 2 shows the compression of the refrigerant vapors in the compressor. The line between points 2 and 3 shows the cooling of the refrigerant vapor in the condenser that takes place until the condensing temperature is reached at the selected pressure. The straight line

between points 3 and 4 indicates the condensation of the refrigerant in the condenser. The line connecting points 4 and 5 indicates subcooling of the condensed refrigerant in the condenser. The condensed refrigerant is subcooled further in the IHE, as shown by the yellow line connecting points 5 and 6. The line between 6 and 7 indicates the expansion of the condensed refrigerant. Evaporation of the refrigerant in the evaporator is shown by the line connecting points 7 and 8. The superheating of the refrigerant in the evaporator is represented by the line between points 8 and 9. The yellow line between points 9 and 1 represents the superheating of the refrigerant in the IHE.

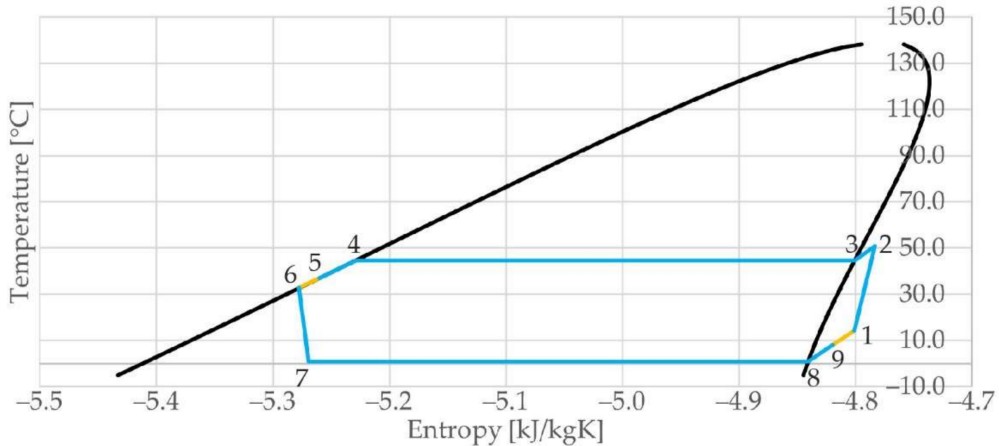

**Figure 8.** T–S diagram of heat pump with an IHE when using R1336mzz(E) as the refrigerant.

Figure 9 shows the dependence of the COP value of the heat pump on the outflow temperature of the heated medium when different refrigerants are used. The graph shows the curve for R1336mzz(E), where the isentropic efficiency of the compressor was equal to 80%. The COP values were similar when different refrigerants were used. The highest values at all three different outlet temperatures of the heated medium were reached by R1336mzz(E). At the same isentropic efficiencies, the use of R1336mzz(E) achieved about 1.6% higher COP values than the use of R134a at all three different outlet temperatures. Using R1234ze(E) achieved between 1.4% and 5.3% lower COP values than using R1336mzz(E). The higher the desired outlet temperature of the heated water, the greater the difference between the COP value when using R1336mzz(E) and the COP value when using R1234ze(E). In practice, the isentropic efficiency of the compressor is lower due to the superheating of vapor in the IHE, and consequently, the COP of the heat pump is lower.

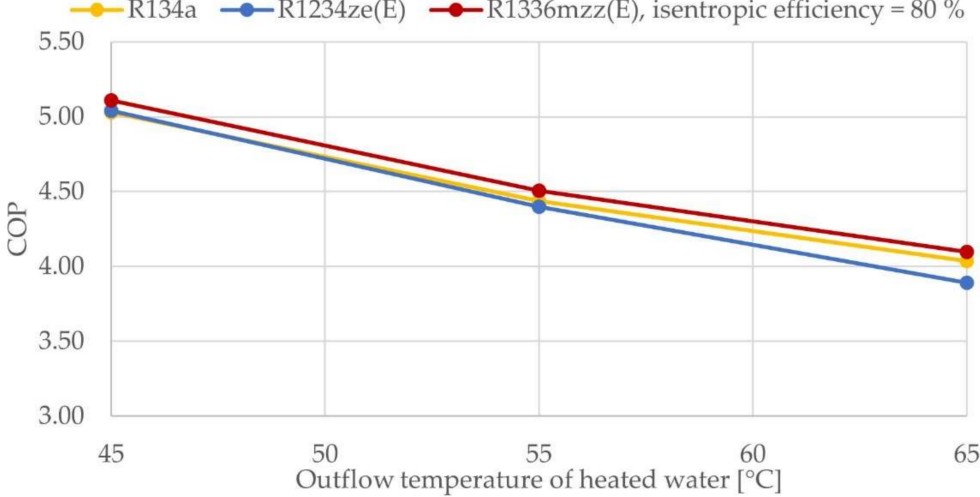

**Figure 9.** Effect of the outlet temperature of the heated water on the COP.

The value COP decreased the most when using the refrigerant R1234ze(E) as the outlet temperature of the heated water increased. When the outlet temperature of the heated water increased from 45 to 65 °C, the COP value of the heat pump decreased by 22.8% if the refrigerant R1234ze(E) was used. When R134a and R1336mzz(E) were used, the COP value decreased by 19.8% for the same change in outlet temperature.

As mentioned earlier, superheating of the refrigerant vapor reduces the isentropic efficiency of the compressor. Since the superheating of the vapor is mandatory when using the refrigerant R1336mzz(E), an analysis was also performed of the effect of the decrease in isentropic efficiency on the value COP. Figure 10 shows the correlation between the value of COP of the heat pump and the isentropic efficiency of the compressor. It was found that, when the isentropic efficiency decreased by a certain percentage, the COP value of the heat pump decreased by approximately the same percentage. In the simulated cases, when the isentropic efficiency was reduced by 20%, the COP value decreased by about 20% at all three different outflow temperatures of the heated water.

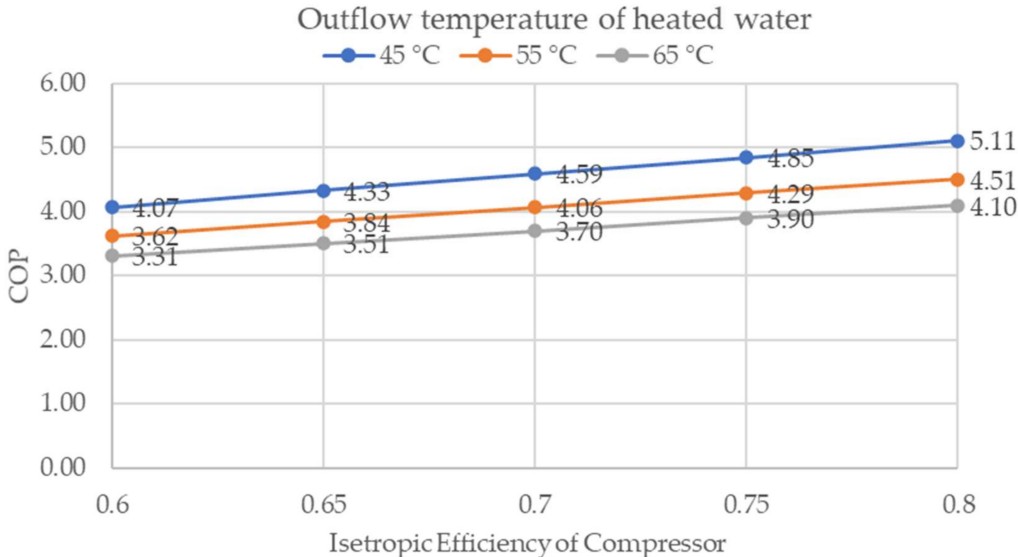

**Figure 10.** Effect of superheating of the refrigerant vapor in the IHE on the COP of the heat pump.

A comparison of different heat pump systems for heating to 55 and 65 °C was simulated using the refrigerants R134a and R1234ze(E). The refrigerant R1336mzz(E) was not used in these comparisons because its use requires superheating with an IHE at a given heat source.

Figure 11 shows a comparison of the COP values in the cases where the water was heated to 55 °C with a heat pump, when the water was heated to 55 °C with a series of two heat pumps where the condensers were connected, and with a series of two heat pumps where the evaporators were also connected.

When the refrigerant R1234ze(E) was used, almost identical COP values were obtained in all three different heating systems. When R134a was used, the heat pump system where the evaporators of the heat pumps were also connected achieved a 2.6% higher COP.

The values for the mass flows, temperatures of the streams, pressures, heat duty and work of the compressor when using both refrigerants can be found in Appendix A.

Figure 12 shows a comparison of the COP values in the cases where the water was heated to 65 °C with a heat pump, when the water was heated to 65 °C with a series of heat pumps where three condensers were connected (the scheme in Figure 3), and with a series of heat pumps where the three evaporators were also connected (the scheme in Figure 4).

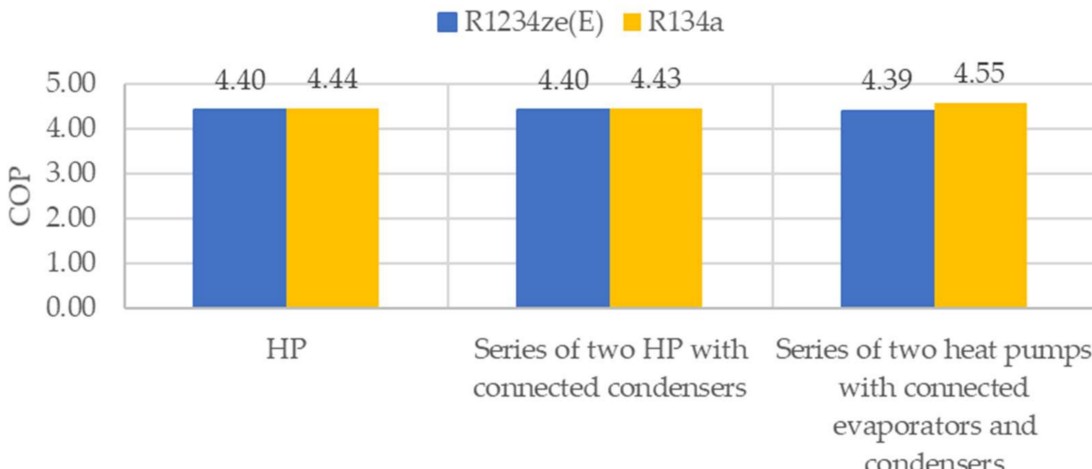

**Figure 11.** Comparison of COP values of different heat pump systems for hot water heating at 55 °C.

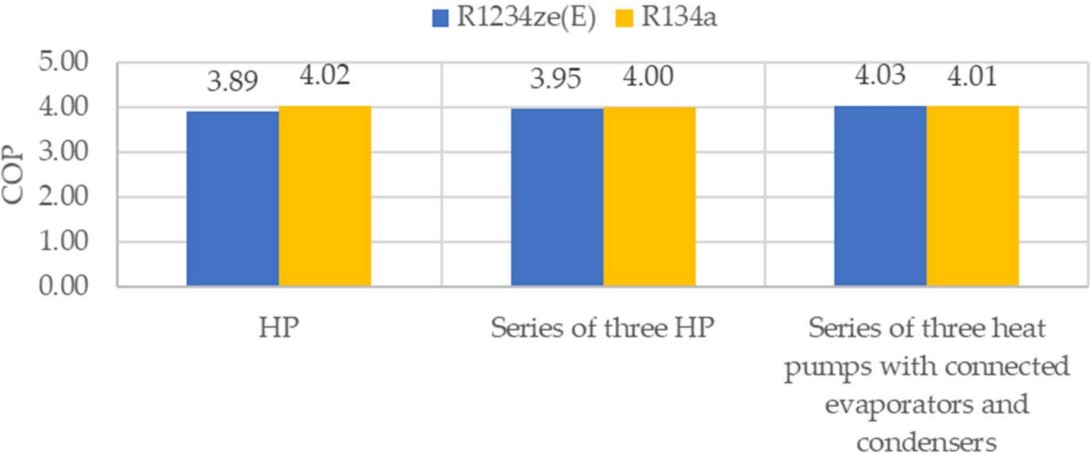

**Figure 12.** Comparison of COP values of different heat pump systems for hot water heating at 65 °C.

In this case, when the refrigerant R134a was used, almost identical COP values were obtained in all three systems. When the refrigerant R1234ze(E) was used, the COP value improved by 1.4% when the heat pump condensers were connected together. In a system where the heat pump evaporators were also connected together, the COP value was 3.5% higher than when the water was heated from 35 to 65 °C with a single heat pump.

In the system where condensers and evaporators were connected together, a much higher mass flow rate of the heat source had to be provided when R1234ze(E) was used. This was necessary because the vapors of R1234ze(E) begin to condense in the compressor when the inlet temperatures are too low. With the higher mass flow rate of the heat source, we ensured a sufficiently high temperature of the saturated R1234ze(E) vapors such that condensation did not occur in the compressor. As a result, the temperature of the heat source outflow ($T_{S,OUT,C}$) was much higher in this system than in the other cases.

To prevent condensation of R1234ze(E) refrigerant vapors in the compressor, it was necessary to adjust the outlet pressure at the expansion valve ($p_{EV,A}$ in $p_{EV,B}$). If this was too high, the R1234ze(E) vapors condensed in the compressor, regardless of how high the compressor discharge pressure was ($p_{COMP,A}$ in $p_{COMP,B}$).

The values for the mass flows, temperature of the flows, pressures, heat and work of the compressor with using both refrigerants can be found in Appendix B.

## 4. Conclusions

It has been shown that R1336mzz(E) can replace R134a, but not as well as R1234ze(E), because the use of R1336mzz(E) requires much more superheating of the refrigerant vapor before it enters the compressor, which affects the isentropic efficiency of the heat pump compressor negatively. If the isentropic efficiencies were the same for all three refrigerants, the highest COP values were obtained with R1336mzz(E) at all three different outlet temperatures of the heated water.

For a given heat source, it is essential to include the IHE in the heat pump system when using the R1336mzz(E) refrigerant. Otherwise, there will be a temperature crossover in the evaporator of the heat pump. For the same isentropic efficiency, higher COP values were obtained with R1336mzz(E) than with the R134a and R1234ze(E) refrigerants.

During the investigation, it was found that if the isentropic efficiency of the compressor was reduced by one percent, the COP value of the heat pump decreased by one percent. Therefore, in the future, it would be necessary to conduct a laboratory study, the result of which would be an equation describing the change in isentropic efficiency of the compressor when superheated saturated vapors enter the compressor.

When comparing different heat pump systems in heating water for higher temperature differences, it was found that the values of COP in the case where only the heat pump condensers were connected showed no difference from the case where the water was heated by only one heat pump. Higher COP values were usually obtained when the evaporators of the heat pumps were connected after the condensers. When using the classical refrigerant R134a, the flows of the heated medium were similar in all three systems, and when using the newer refrigerant R1234ze(E), it was necessary to ensure higher heat source flows in the system when the evaporators of the heat pumps were connected; otherwise, condensation of the saturated refrigerant vapors occurred in the compressor of the second or third heat pumps.

**Author Contributions:** J.D.: Methodology, Formal analysis, Investigation, Writing—Original Draft preparation. D.U.: Investigation, Supervision. D.G.: Conceptualization, Validation, Supervision. All authors have read and agreed to the published version of the manuscript.

**Funding:** Research was funded by ARRS (Slovenian Research Agency) research and infrastructure programs P2-0046 Separation Processes and Product Design and P2-0414 Process Systems Engineering and Sustainable Development.

**Institutional Review Board Statement:** Not applicable.

**Informed Consent Statement:** Not applicable.

**Data Availability Statement:** All the data used in this study are available in the text, graphs and tables of this article.

**Conflicts of Interest:** The authors declare no conflict of interest.

## Nomenclature

*Symbols*

| | |
|---|---|
| $COP$ | coefficient of performance (/) |
| $p_{COMP,i}$ | absolute discharge pressure of the compressor of the i heat pump in series (bar) |
| $p_{EV,i}$ | absolute outlet pressure of the expansion valve of the i heat pump in series (bar) |
| $qm_{HW}$ | mass flow of the heated medium through the system (kg/h) |
| $qm_{R,i}$ | mass flow of the refrigerant through the i heat pump in series (kg/h) |
| $qm_R$ | sum of mass flows of the refrigerant through all heat pumps in series (kg/h) |
| $qm_{S,i}$ | mass flow of the heat source through the evaporator of the i heat pump in series (kg/h) |

| $qm_S$ | sum of mass flows of the heat source through evaporators of all heat pumps in series (kg/h) |
| $Q_{COND,i}$ | heat duty of the condenser of the i heat pump in series (kW) |
| $Q_{EVAP,i}$ | heat duty of the evaporator of the i heat pump in series (kW) |
| $S$ | entropy (kJ/kgK) |
| $T_{1,i}$ | temperature of the flow from the evaporator to the compressor in the i heat pump in series (°C) |
| $T_{1,A,MIN}$ | minimum vapor inlet temperature into the compressor (°C) |
| $T_{2,i}$ | temperature of the flow from the compressor to the condenser in the i heat pump in series (°C) |
| $T_{3,i}$ | temperature of the flow from the condenser to the expansion valve in the i heat pump in series (°C) |
| $T_{4,i}$ | temperature of the flow from the expansion valve to the evaporator in the first heat pump in series (°C) |
| $T_{HW,AB}$ | temperature of the flow of the heated medium from the condenser of the first heat pump in series to the condenser of the second heat pump in series (°C) |
| $T_{HW,BC}$ | temperature of the flow of the heated medium from the condenser of the second heat pump in series to the condenser of the third heat pump in series (°C) |
| $T_{HW,IN}$ | temperature of the inflow of the heated medium into the condenser of the first heat pump in series (°C) |
| $T_{HW,OUT}$ | temperature of the outflow of the heated medium from the condenser of the third heat pump in series (°C) |
| $T_{S,IN,i}$ | temperature of the inflow of the heat source into the evaporator of the i heat pump in series (°C) |
| $T_{S,OUT,i}$ | temperature of the outflow of the heat source from the evaporator of the i heat pump in series (°C) |
| $W_{actual}$ | work of the real compression process (kW) |
| $W_{COMP,i}$ | work of the compressor of the first heat pump in series (kW) |
| $W_{isen}$ | work of the isentropic compression process (kW) |
| ***Substract i in symbols*** | |
| A | the first heat pump in series |
| B | the second heat pump in series |
| C | the third heat pump in series |
| ***Greek Symbols*** | |
| $\eta_{isen}$ | compressor efficiency |

## Abbreviations

| ASHRAE | The American Society of Heating, Refrigerating and Air-Conditioning Engineers |
| CFC | chlorofluorocarbons |
| COP | coefficient of performance |
| GHG | greenhouse gas |
| GWP | global warming potential |
| HCFC | hydrochlorofluorocarbons |
| HFC | hydrofluorocarbons |
| HFO | hydrofluoroolefins |
| HP | heat pump |
| IHE | internal heat exchanger |
| NBP | normal boiling point |
| ODP | ozone depletion potential |

## IUPAC Names of Refrigerants

| R1234yf | 2,3,3,3-tetrafluoroproene |
| R1234ze(E) | trans-1,3,3,3-tetrafluoropropene |
| R1336mzz(E) | trans-1,1,1,4,4,4-hexafluoro-2-butene |

| R134a | 1,1,1,2-tetrafluoroethane |
|---|---|
| R152a | 1,1-difluoroethane |
| R450A | mixture of R1234ze(E) and R134a |
| R515B | mixture of R1234ze(E) and R227ea |

## Appendix A

**Table A1.** The values for the mass flows, temperatures of the streams, pressures, heat duty and work of the compressor of different heat pump systems for hot water heating at 55 °C.

| R134a | HP | HP with Connected CONDENSERS | $p_{EV}$ HP with Connected CONDENSERS and EVAPORATORS |
|---|---|---|---|
| $qm_{R,A}$ (kg/h) | 500 | 282 | 287 |
| $qm_{R,B}$ (kg/h) | | 248 | 252 |
| $qm_R$ (kg/h) | 500 | 530 | 539 |
| $qm_{S,IN,A}$ (kg/h) | 2835 | 1470 | 2800 |
| $qm_{S,IN,B}$ (kg/h) | | 1130 | |
| $qm_{S,IN}$ (kg/h) | 2835 | 2600 | 2800 |
| $qm_{HW,IN}$ (kg/h) | 1090 | 1089 | 1090 |
| $T_{1,A}$ (°C) | 8 | 8 | 8 |
| $T_{2,A}$ (°C) | 72.09 | 63.71 | 60.83 |
| $T_{3,A}$ (°C) | 37.02 | 37.07 | 37.15 |
| $T_{4,A}$ (°C) | 1 | 1 | 3.46 |
| $T_{1,B}$ (°C) | | 8 | 4.51 |
| $T_{2,B}$ (°C) | | 73.06 | 69.89 |
| $T_{3,B}$ (°C) | | 48.84 | 48.36 |
| $T_{4,B}$ (°C) | | 1 | 1 |
| $T_{S,IN,A}$ (°C) | 10 | 10 | 10 |
| $T_{S,OUT,A}$ (°C) | 3.97 | 3.45 | 6.51 |
| $T_{S,IN,B}$ (°C) | | 10 | 6.51 |
| $T_{S,OUT,B}$ (°C) | | 3.41 | 3.86 |
| $T_{HW,IN}$ (°C) | 35 | 35 | 35 |
| $T_{HW,AB}$ (°C) | | 46 | 46 |
| $T_{HW,OUT}$ (°C) | 55 | 55 | 55 |
| $p_{COMP,A}$ (bar) | 14.3 | 11.9 | 12 |
| $p_{EV,A}$ (bar) | 3.025 | 3.025 | 3.302 |
| $p_{COMP,B}$ (bar) | | 14.6 | 14.7 |
| $p_{EV,B}$ (bar) | | 3.025 | 3.025 |
| $Q_{EVAP,A}$ (kW) | 21.538 | 12.134 | 12.302 |
| $Q_{EVAP,B}$ (kW) | | 9.378 | 9.382 |
| $Q_{EVAP}$ (kW) | 21.538 | 21.512 | 21.684 |
| $Q_{COND,A}$ (kW) | 27.404 | 15.052 | 15.066 |
| $Q_{COND,B}$ (kW) | | 12.326 | 12.338 |
| $Q_{COND}$ (kW) | 27.404 | 27.378 | 27.404 |
| $W_{COMP,A}$ (kW) | 6.175 | 3.072 | 2.909 |
| $W_{COMP,B}$ (kW) | | 3.102 | 3.111 |
| $W_{COMP}$ (kW) | 6.175 | 6.174 | 6.020 |
| *COP* | 4.44 | 4.43 | 4.55 |

| R1234ze(E) | HP | HP with connected CONDENSERS | $p_{EV}$ HP with connected CONDENSERS and EVAPORATORS |
|---|---|---|---|
| $qm_{R,A}$ (kg/h) | 500 | 282 | 285 |
| $qm_{R,B}$ (kg/h) | | 248 | 250 |
| $qm_R$ (kg/h) | 500 | 530 | 535 |
| $qm_{S,IN,A}$ (kg/h) | 2615 | 1470 | 3075 |
| $qm_{S,IN,B}$ (kg/h) | | 1130 | |
| $qm_{S,IN}$ (kg/h) | 2615 | 2600 | 3075 |
| $qm_{HW,IN}$ (kg/h) | 1007 | 1009 | 1090 |

**Table A1.** *Cont.*

| | | | | |
|---|---|---|---|---|
| $T_{1,A}$ (°C) | 8 | 8 | | 8 |
| $T_{2,A}$ (°C) | 64.89 | 57.93 | | 57.28 |
| $T_{3,A}$ (°C) | 37.11 | 37.09 | | 37.13 |
| $T_{4,A}$ (°C) | 1 | 1 | | 4.02 |
| $T_{1,B}$ (°C) | | 8 | | 5.09 |
| $T_{2,B}$ (°C) | | 65.25 | | 65.14 |
| $T_{3,B}$ (°C) | | 48.15 | | 48.08 |
| $T_{4,B}$ (°C) | | 1 | | 1 |
| $T_{S,IN,A}$ (°C) | 10 | 10 | | 10 |
| $T_{S,OUT,A}$ (°C) | 3.97 | 3.95 | | 7.09 |
| $T_{S,IN,B}$ (°C) | | 10 | | 7.09 |
| $T_{S,OUT,B}$ (°C) | | 3.89 | | 4.86 |
| $T_{HW,IN}$ (°C) | 35 | 35 | | 35 |
| $T_{HW,AB}$ (°C) | | 46 | | 46 |
| $T_{HW,OUT}$ (°C) | 55 | 55 | | 55 |
| $p_{COMP,A}$ (bar) | 11.3 | 9.3 | | 10 |
| $p_{EV,A}$ (bar) | 2.206 | 2.206 | | 2.462 |
| $p_{COMP,B}$ (bar) | | 11.2 | | 12 |
| $p_{EV,B}$ (bar) | | 2.206 | | 2.206 |
| $Q_{EVAP,A}$ (kW) | 19.852 | 11.196 | | 11.277 |
| $Q_{EVAP,B}$ (kW) | | 8.696 | | 8.601 |
| $Q_{EVAP}$ (kW) | 19.852 | 19.892 | | 19.878 |
| $Q_{COND,A}$ (kW) | 25.317 | 13.950 | | 13.947 |
| $Q_{COND,B}$ (kW) | | 11.421 | | 11.421 |
| $Q_{COND}$ (kW) | 25.317 | 25.368 | | 25.368 |
| $W_{COMP,A}$ (kW) | 5.753 | 2.895 | | 2.810 |
| $W_{COMP,B}$ (kW) | | 2.869 | | 2.968 |
| $W_{COMP}$ (kW) | 5.753 | 5.764 | | 5.778 |
| *COP* | 4.40 | 4.40 | | 4.39 |

## Appendix B

**Table A2.** The values for the mass flows, temperatures of the streams, pressures, heat duty and work of the compressor of different heat pump systems for hot water heating at 65 °C.

| R134a | HP | HP with Connected CONDENSERS | $p_{EV}$ HP with Connected CONDENSERS and EVAPORATORS |
|---|---|---|---|
| $qm_{R,A}$ (kg/h) | 500 | 212 | 213 |
| $qm_{R,B}$ (kg/h) | | 189 | 194 |
| $qm_{R,C}$ (kg/h) | | 162 | 167 |
| $qm_R$ (kg/h) | 500 | 563 | 574 |
| $qm_{S,IN,A}$ (kg/h) | 2515 | 1300 | 2820 |
| $qm_{S,IN,B}$ (kg/h) | | 1000 | |
| $qm_{S,IN,C}$ (kg/h) | | 700 | |
| $qm_{S,IN}$ (kg/h) | 2515 | 3000 | 2820 |
| $qm_{HW,IN}$ (kg/h) | 747 | 750 | 750 |
| $T_{1,A}$ (°C) | 8 | 8 | 8 |
| $T_{2,A}$ (°C) | 82.25 | 63.71 | 66.32 |
| $T_{3,A}$ (°C) | 37.13 | 37.14 | 37.51 |
| $T_{4,A}$ (°C) | 1 | 1 | 4.43 |
| $T_{1,B}$ (°C) | | 8 | 5.44 |
| $T_{2,B}$ (°C) | | 75.26 | 71.13 |

**Table A2.** *Cont.*

| | | | |
|---|---|---|---|
| $T_{3,B}$ (°C) | | 49.16 | 49.22 |
| $T_{4,B}$ (°C) | | 1 | 2.47 |
| $T_{1,C}$ (°C) | | 8 | 3.42 |
| $T_{2,C}$ (°C) | | 84.68 | 80.21 |
| $T_{3,C}$ (°C) | | 59.12 | 59.13 |
| $T_{4,C}$ (°C) | | 1 | 1 |
| $T_{S,IN,A}$ (°C) | 10 | 10 | 10 |
| $T_{S,OUT,A}$ (°C) | 3.2 | 4.43 | 7.44 |
| $T_{S,IN,B}$ (°C) | | 10 | 7.44 |
| $T_{S,OUT,B}$ (°C) | | 4.35 | 5.42 |
| $T_{S,IN,C}$ (°C) | | 10 | 5.42 |
| $T_{S,OUT,C}$ (°C) | | 3.94 | 3.92 |
| $T_{HW,IN}$ (°C) | 35 | 35 | 35 |
| $T_{HW,AB}$ (°C) | | 47 | 47 |
| $T_{HW,BC}$ (°C) | | 57 | 57 |
| $T_{HW,OUT}$ (°C) | 65 | 65 | 65 |
| $p_{COMP,A}$ (bar) | 17.7 | 11.9 | 13.9 |
| $p_{EV,A}$ (bar) | 3.025 | 3.025 | 3.42 |
| $p_{COMP,B}$ (bar) | | 15.3 | 15.4 |
| $p_{EV,B}$ (bar) | | 3.025 | 3.187 |
| $p_{COMP,C}$ (bar) | | 18.6 | 18.6 |
| $p_{EV,C}$ (bar) | | 3.025 | 3.025 |
| $Q_{EVAP,A}$ (kW) | 21.416 | 9.155 | 9.091 |
| $Q_{EVAP,B}$ (kW) | | 7.122 | 7.175 |
| $Q_{EVAP,C}$ (kW) | | 5.343 | 5.329 |
| $Q_{EVAP}$ (kW) | 21.416 | 21.580 | 21.595 |
| $Q_{COND,A}$ (kW) | 28.190 | 11.309 | 11.309 |
| $Q_{COND,B}$ (kW) | | 9.434 | 9.434 |
| $Q_{COND,C}$ (kW) | | 7.559 | 7.559 |
| $Q_{COND}$ (kW) | 28.190 | 28.302 | 28.302 |
| $W_{COMP,A}$ (kW) | 7.008 | 2.310 | 2.335 |
| $W_{COMP,B}$ (kW) | | 2.434 | 2.378 |
| $W_{COMP,C}$ (kW) | | 2.333 | 2.348 |
| $W_{COMP}$ (kW) | 7.008 | 7.077 | 7.061 |
| *COP* | 4.02 | 4.00 | 4.01 |

| R1234ze(E) | HP | HP with connected CONDENSERS | $p_{EV}$ HP with connected CONDENSERS and EVAPORATORS |
|---|---|---|---|
| $qm_{R,A}$ (kg/h) | 500 | 212 | 215 |
| $qm_{R,B}$ (kg/h) | | 191 | 193 |
| $qm_{R,C}$ (kg/h) | | 164 | 166 |
| $qm_R$ (kg/h) | 500 | 567 | 574 |
| $qm_{S,IN,A}$ (kg/h) | 2615 | 1200 | 4249 |
| $qm_{S,IN,B}$ (kg/h) | | 900 | |
| $qm_{S,IN,C}$ (kg/h) | | 650 | |
| $qm_{S,IN}$ (kg/h) | 2615 | 2750 | 4249 |
| $qm_{HW,IN}$ (kg/h) | 697 | 697 | 693 |
| $T_{1,A}$ (°C) | 8 | 8 | 8 |
| $T_{2,A}$ (°C) | 77.18 | 58.75 | 56.68 |
| $T_{3,A}$ (°C) | 37.02 | 37.14 | 37.52 |
| $T_{4,A}$ (°C) | 1 | 1 | 4.55 |
| $T_{1,B}$ (°C) | | 8 | 6.35 |
| $T_{2,B}$ (°C) | | 67.36 | 64.84 |
| $T_{3,B}$ (°C) | | 49.30 | 49.07 |
| $T_{4,B}$ (°C) | | 1 | 4.20 |

Table A2. *Cont.*

| | | | |
|---|---|---|---|
| $T_{1,C}$ (°C) | | 8 | 5.19 |
| $T_{2,C}$ (°C) | | 75.70 | 72.23 |
| $T_{3,C}$ (°C) | | 59.14 | 59.03 |
| $T_{4,C}$ (°C) | | 1 | 1 |
| $T_{S,IN,A}$ (°C) | 10 | 10 | 10 |
| $T_{S,OUT,A}$ (°C) | 3.96 | 4.43 | 8.42 |
| $T_{S,IN,B}$ (°C) | | 10 | 8.42 |
| $T_{S,OUT,B}$ (°C) | | 4.18 | 7.19 |
| $T_{S,IN,C}$ (°C) | | 10 | 7.19 |
| $T_{S,OUT,C}$ (°C) | | 3.95 | 6.27 |
| $T_{HW,IN}$ (°C) | 35 | 35 | 35 |
| $T_{HW,AB}$ (°C) | | 47 | 47 |
| $T_{HW,BC}$ (°C) | | 57 | 57 |
| $T_{HW,OUT}$ (°C) | 65 | 65 | 65 |
| $p_{COMP,A}$ (bar) | 14.9 | 9.5 | 10 |
| $p_{EV,A}$ (bar) | 2.026 | 2.026 | 2.51 |
| $p_{COMP,B}$ (bar) | | 11.8 | 12 |
| $p_{EV,B}$ (bar) | | 2.026 | 2.32 |
| $p_{COMP,C}$ (bar) | | 14.4 | 14.5 |
| $p_{EV,C}$ (bar) | | 2.026 | 2.206 |
| $Q_{EVAP,A}$ (kW) | 19.883 | 8.412 | 8.468 |
| $Q_{EVAP,B}$ (kW) | | 6.603 | 6.591 |
| $Q_{EVAP,C}$ (kW) | | 4.955 | 4.912 |
| $Q_{EVAP}$ (kW) | 19.883 | 19.970 | 19.971 |
| $Q_{COND,A}$ (kW) | 26.303 | 10.510 | 10.450 |
| $Q_{COND,B}$ (kW) | | 8.768 | 8.717 |
| $Q_{COND,C}$ (kW) | | 7.052 | 6.985 |
| $Q_{COND}$ (kW) | 26.303 | 26.303 | 26.152 |
| $W_{COMP,A}$ (kW) | 6.758 | 2.208 | 2.086 |
| $W_{COMP,B}$ (kW) | | 2.278 | 2.223 |
| $W_{COMP,C}$ (kW) | | 2.179 | 2.182 |
| $W_{COMP}$ (kW) | 6.758 | 6.665 | 6.491 |
| *COP* | 3.89 | 3.95 | 4.03 |

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
