# Peer review of "Comparison of the New Refrigerant R1336mzz(E) with R1234ze(E) as an Alternative to R134a for Use in Heat Pumps"

_processes, doi:10.3390/pr10020218_

Round 1

Reviewer 1 Report

Checking spelling and grammar

Reviewer 2 Report

This manuscript analyzes the performance of R1336mzz(E) and R1234ze(E) replacing R134a in heat pump system. However, this work is not novelly, and there are too few working fluids considered. The authors need to rewrite the paper and add more information.

  1. The title "comparison of new refrigerant R1336mzz (E) and R1234ze (E) as alternatives to R134a".It should be added that the application occasion is heat pump in the title.
  2. R1234yf as an important alternative working fluidof R134a, should also be calculated as a reference. Why not consider R1233zd (E) and R1336mzz(Z), which are nonflammable and suitable for heat pumps. The authors should make a more comprehensive analysis and comparison of various HFOs working fluids replacing R134a under the design condition.
  3. This workdescribes the inlet and outlet temperature of the heat exchanger. Whether there are enough data on thermodynamic properties and transport properties of the considered working fluidsto discuss the heat exchanger in detail, such as the size and investment of the heat exchanger. If the data are not enough, the heat exchanger UA can be calculated and compared.
  4. This work mainly compares the COP of three working fluids in heat pump system. Compared with R134a, the volumetric heating capacity of HFO may be quite different. The authors should also compare the volumetric heating capacity of different working fluids under design conditions.
  5. If achieving the same heating capacity, the mass flow of three working fluids will be different. How will this difference affect the performance of various parts of the heat pump (heat exchanger and compressor). Whether it can directly fillthe HFOs to replace R134a incurrent heat pump system, or it must redesign the system?
  6. Have the authors considered optimization of evaporation temperature and condensation temperature of alternative working fluids?

Reviewer 3 Report

General comments:

Writing English should be improved.

The paper format is good but the authors should improve the content, especially the result and discussion section.

The introduction takes up too much space for the basic knowledge. Thus the manuscript is unnecessarily long.

Reviewer 4 Report

R1336mzz is evaluated as an alternative to R134a, most widely used refrigerant. R1336mzz is one of typical HFOs with low GWP. Hence, it is necessary to investigate the performance with comparison of R134a. It is recommended for publication following minor revision. (1) the performance was evaluated by ASPEN. It will be more consolidated with experimental verification. (2) The effect of outlet temperature of the heated water was investigated on the COP. How about the vapor pressure or compression pressure? (3) COP values of different heat pump systems for hot water heating were evaluated for R134 and R1234ze, rather than R1336. (4) As indicated by the authors, R1234yf is the first HFO tested. Why 1234yf was not included for comparision? (5) Was the investigation verified before simulations to assure the accuracy of the results? (6) Is it possible to evaluate the performance of a mixture containing R1336?

Round 2

Reviewer 2 Report

The authors have made appropriate changes to the vast majority of the review comments. I still suggest that the title should make it clear that the application of this article is a heat pump.

Reviewer 3 Report

The authors have corrected the manuscript accordingly. The manuscript is quite long (I would say it due to the unnecessary parts in sections 2.1 and 2.2). I would recommend the authors to generally improve before it can be published.
